# Systemic Effects of Radiotherapy and Concurrent Chemo-Radiotherapy in Head and Neck Cancer Patients—Comparison of Serum Metabolome Profiles

**DOI:** 10.3390/metabo10020060

**Published:** 2020-02-07

**Authors:** Karol Jelonek, Aleksandra Krzywon, Patrycja Jablonska, Ewa M. Slominska, Ryszard T. Smolenski, Joanna Polanska, Tomasz Rutkowski, Jolanta Mrochem-Kwarciak, Krzysztof Skladowski, Piotr Widlak

**Affiliations:** 1Maria Sklodowska-Curie National Research Institute of Oncology, Gliwice Branch, Wybrzeze Armii Krajowej 15, 44-102 Gliwice, Poland; Aleksandra.Krzywon@io.gliwice.pl (A.K.); Tomasz.Rutkowski@io.gliwice.pl (T.R.); Jolanta.Mrochem-Kwarciak@io.gliwice.pl (J.M.-K.); Krzysztof.Skladowski@io.gliwice.pl (K.S.); Piotr.Widlak@io.gliwice.pl (P.W.); 2Department of Biochemistry, Medical University of Gdansk, Debinki 1, 80-211 Gdansk, Poland; patrycja.s@gumed.edu.pl (P.J.); eslom@gumed.edu.pl (E.M.S.); rt.smolenski@gumed.edu.pl (R.T.S.); 3Department of Data Science and Engineering, Silesian University of Technology, Akademicka 16, 44-100 Gliwice, Poland; Joanna.Polanska@polsl.pl

**Keywords:** head and neck cancer, mass spectrometry, metabolomics, radiation response, chemotherapy response

## Abstract

Anticancer treatment induces systemic molecular changes that could be detected at the level of biofluids. Understanding how human metabolism is influenced by these treatments is crucial to predict the individual response and adjust personalized therapies. Here, we aimed to compare profiles of metabolites in serum of head and neck cancer patients treated with concurrent chemo-radiotherapy, radiotherapy alone, or induction chemotherapy. Serum samples were analyzed by a targeted quantitative approach using combined direct flow injection and liquid chromatography coupled to tandem mass spectrometry, which allowed simultaneous quantification of 149 metabolites. There were 45 metabolites whose levels were significantly changed between pretreatment and within- or post-treatment serum samples, including 38 phospholipids. Concurrent chemo-radiotherapy induced faster and stronger effects than radiotherapy alone. On the other hand, chemotherapy alone did not result in significant changes. The decreased level of total phospholipids was the most apparent effect observed during the first step of the treatment. This corresponded to the loss of patients’ body mass, yet no correlation between both parameters was observed for individual patients. We concluded that different molecular changes were measured at the level of serum metabolome in response to different treatment modalities.

## 1. Introduction

The primary goal of every cancer treatment is to eliminate as many tumor cells as possible. However, preserving the function of the adjacent normal tissues is equally important, which poses a real challenge for treatment planning [1]. The treatment of head and neck squamous cell cancers (HNSCC) presents many challenges in particular since this region contains many critical structures that could be affected by tumor treatment or the tumor itself. Major treatment options next to surgery are chemotherapy and radiation therapy. Chemotherapy is a systemic treatment that utilizes the drug(s) to destroy cancer cells or prevent their growth [2]. Radiation therapy (RT), on the other hand, is the use of high-energy ionizing radiation to destroy cancer cells locally [3]. In order to improve the efficacy of the treatment, chemo- and radiotherapy can also be combined, delivering more aggressive cancer treatment. In this case, normal tissue reactions might significantly influence the patient’s quality of life. Therefore, the prediction of adverse effects of the treatment before or at the early stages of therapy is an important issue of personalized medicine. 

One of the approaches to assess the systemic response of the patient to treatment is to monitor molecular markers in biofluids. Among all omics platforms, metabolomics has great potential to impact clinical practices due to its ability to rapidly analyze samples with little sample preparation and enough sensitivity to reflect the influence of different internal and external stimuli [4]. Compared with genomics or proteomics, metabolomics directly reflects changes in phenotype and function, which is complementary to “upstream” changes observed at the level of transcripts and proteins [5]. The response of the human metabolome to radiotherapy assessed in biofluids is still only marginally addressed in the literature. Available reports show radiation-induced changes in metabolite profiles in urine [6] and in red blood cells [7]. Blood-based studies show changes in patients with HNSCC caused by RT in serum phospholipids [8] as well as in a broader spectrum of small metabolites [9]. Changes in the metabolic composition of blood were also associated with the toxicity of RT, and a detailed review regarding this correlation as well as systemic effects of ionizing radiation at the serum/plasma metabolome level is presented elsewhere [10]. On the other hand, chemotherapy-induced alterations in the circulating metabolome were only reported in one study based on serum from patients treated due to metastatic renal cell carcinoma [11]. However, there is no metabolomics data available regarding effects of chemo-radiotherapy and their comparison with unimodal treatments. Therefore, in this study, we aimed to detect and compare changes induced at the serum metabolome level by three modalities of anticancer treatment: radiotherapy or chemotherapy alone and concurrent chemo-radiotherapy. Metabolite profiles were analyzed by mass spectrometry in sera collected from patients treated due to HNSCC before, during, and after the end of therapy. 

## 2. Results

To compare the effects of different modalities of cytotoxic anticancer therapy on the serum metabolome profile, three groups of patients treated due to locally advanced HNSCC were included. Two groups were treated with schemes involving ionizing radiation: radiotherapy alone (RT) and concurrent chemo-radiotherapy (CCRT) involving cisplatin; samples were collected before (sample A), about 2 weeks after the start (sample B), and just after the end (sample C) of a treatment. Chemotherapy alone is not a standard in HNSCC treatment, hence, a group treated with induction chemotherapy (ICT) was included and samples were collected before (sample A) and after (sample B) the treatment with cisplatin or cisplatin and 5-fluorouracil; sample B in this group corresponded to within-treatment sample B in the CCRT group regarding time and dose of administered drug (Figure 1).

In general, there were 149 metabolites detected and quantified in analyzed samples. Using the Principal component analysis (PCA) analysis, we compared the overall similarity between groups. The analysis showed an overlap of pretreatment samples A from all three groups. However, a separation of ICT group and radiation-involving groups was noted for samples B, especially considering the CCRT group (Figure 2A). Then, we analyzed both radiation-involved groups at all time points. We found clear separation of specimens collected after the end of a treatment (sample C), yet differences between both groups of patients were also visible in specimens collected about 2 weeks after start of a treatment (sample B); substantial overlap of specimens collected before a treatment (point A) was retained (Figure 2B). 

To search for specific metabolites whose abundances changed between subsequent time points, we estimated the significance of individual changes between consecutive samples from the same patient, which reflected the significance of treatment-induced changes in a putatively heterogeneous group of patients. All serum metabolites that changed their abundance significantly (false discovery rate (FDR)<0.05) between pretreatment sample A and within-treatment sample B or between within-treatment sample B and post-treatment sample C in either group with the radiotherapy involved (i.e., RT and CCRT) are listed in Table 1. 

It is important to note that, in the ICT group, no metabolite changed its abundance significantly between pre- and post-treatment samples (B/A ratio). The quantitation was based on the Biocrates Absolute IDQ p180 platform. It is noteworthy that significant treatment-related changes were noted for the large fraction of detected lipids: 78% of lysoPCs, 50% of sphingomyelins (SMs), and 42% of phosphatidylcholine (PCs) were affected at either time point. On the other hand, a few small metabolites were affected: 19% of biogenic amines, 17% of acylcarnitines, and 14% of amino acids measured by this assay were affected at any time after either RT or CCRT. Moreover, when complementary LC-MS assay for amino acids and their derivatives was implemented, none of the measured compounds showed statistically significant differences between pretreatment and within/post-treatment serum samples (Appendix A). Therefore, we concluded that among targeted metabolites, phospholipids appeared to be significantly altered compared with others.

We found that 31 and 7 metabolites decreased their levels significantly between pre- and within-treatment samples (the B/A ratio) in CCRT and RT groups, respectively. Moreover, 5 metabolites (phosphatidylcholines, PCs) decreased their levels in both groups, yet treatment-induced changes were stronger in the CCRT group (Table 2). We also found that total levels of PC and lysoPC detected in serum were significantly decreased after the first stage of CCRT. Reduced total levels of PC and lysoPC were also noted in samples collected about two weeks after the start of RT alone, yet treatment-induced changes were weaker and did not reach the level of statistical significance (Figure 3). Furthermore, several metabolites changed their abundance significantly between post- and within-treatment samples (the C/B ratio) in both radiation-involving groups. There were 9 and 11 metabolites that changed their abundance in CCRT and RT groups, respectively, including 2 sphingomyelins (SMs) that increased abundance between points B and C in both groups of patients (Table 2). Moreover, total levels of PC and lysoPC increased between points B and C in both groups of patients (Figure 3), yet differences were not statistically significant. Finally, we looked for a difference between pre- and post-treatment samples (the C/A ratio). There were 30 and 23 metabolites that changed their abundance significantly in CCRT and RT, respectively, including 14 metabolites common to both groups (Table 2).

Obtained results indicated differences in patterns of therapy-induced changes between RT and CCRT groups of patients. To address this issue, we defined specific patterns of changes A vs. B vs. C assuming the threshold for statistical significance as FDR < 0.05. Corresponding patterns of changes are presented in Table 1. The analysis revealed 6 metabolites with the same pattern of changes in CCRT and RT. Four metabolites (namely, PC diacyl (aa) C36:3, PC acyl-alkyl (ae) C36:3, PC ae C38:4, and PC ae C38:5) were significantly decreased between points A and B then remained not affected—pattern A > B = C; a similar pattern was observed in the case of 24 and 3 metabolites specific to CCRT and RT groups, respectively. Another 2 metabolites with the same pattern of changes in CCRT and RT groups (namely, SM (OH) C16:1 and SM C18:0) were not affected between points A and B then increased significantly in point C—pattern A = B<C; a similar pattern was observed in case of 4 and 1 metabolites specific to CCRT and RT, respectively. The pattern of treatment-induced changes that were unique for the CCRT group corresponded to decreased level in within-treatment sample B then increased level in post-treatment sample C—pattern A > B < C; this pattern was observed for PC aa 34:2, PC aa C32:3, and PC ae C34:3. It is noteworthy that among metabolites significantly affected in both RT and CCRT groups, there were 3 compounds showing apparent “delay” of changes in RT (pattern A = B > C) compared to CCRT group (pattern A > B = C): kynurenine, PC aa C38:3, and PC aa 40:5. Another metabolite that shows different patterns of change, PC aa C34:2, significantly decreased during the first stage of treatment in both CCRT and RT; yet, at the end of treatment, it increased to the initial pretreatment level only in case of the CCRT group.

To directly compare effects induced by two radiation-involving modalities, the significance of differences between CCRT and RT groups in changes between pre- and within-treatment samples (A vs. B) was estimated. There were two phospholipids where such differences were statistically significant (FDR < 0.05): PC ae C40:3 and lysoPC a C20:3. The abundance of both lipids decreased in samples B in the CCRT group while they remained unchanged in the RT group. On the other hand, abundances of both lipids decreased only in samples C in the RT group (Figure 4), which apparently fits the “delayed response” pattern in the RT group.

It is noteworthy that either RT or CCRT resulted in a decreased concentration of serum albumin (no significant changes were observed in the ICT group). However, the decrease in albumin level was stronger and “faster” in the CCRT group (pattern A > B = C) when compared to the RT group (pattern A = B > C) (Figure 5A), which resembled patterns observed for several metabolites. Moreover, we observed a constant decrease of the body mass during the treatment in both patients’ groups (Figure 5B); the effect was stronger in the CCRT group, particularly in the second stage of the treatment (C-B). Patterns of changes in both serum albumin and body mass were in contrast to the total serum PCs and lysoPCs levels, which did not decrease in the second stage of treatment (Figure 3). 

Noteworthy, although in the first stage of treatment (B-A) the decrease of body mass corresponded to the general decrease of serum phospholipids, no statistically significant correlation was observed between both parameters for individual patients (Figure 6A). However, a weak correlation was observed in the CCRT group between individual changes in serum levels of total phospholipids and albumin (Figure 6B). On the other hand, a weak negative correlation was observed between individual changes in body mass and serum albumin levels in the RT group (Figure 6C). Hence, therapy-related changes in serum levels of phospholipids (and albumin) were not associated directly with a treatment-related drop in the patient’s body mass.

Finally, we compared early toxicity (acute radiation sequel, ARS) that was assessed in a subgroup of patients treated with either CCRT and RT. Similar low toxicity was noted in both patient groups 2 weeks after the start of treatment (point B). However, generally higher acute toxicity was observed in the CCRT group after the end of a treatment (point C), yet differences were not statistically significant due to the small size of compared groups (Figure 7).

## 3. Discussion

Ionizing radiation, either alone or in combination with chemotherapeutic agents, is successfully implemented in the treatment of HNSCC, yet detrimental effects of therapy impairing the patient’s quality of life cannot be neglected. Therefore, molecular factors enabling us to monitor and predict the individual response will help to personalize the treatment. Toxic effects of therapy involve systemic changes that could be detected at the level of serum/plasma proteome and metabolome [10]. Here, we aimed to analyze changes in profiles of serum metabolites induced by two treatment modalities with putatively different toxicity—concurrent chemoradiotherapy (CCRT) and radiotherapy alone (RT). To the best of our knowledge, this is the first study that compares changes in the serum metabolome profiles of HNSCC patients induced by these two modalities (and chemotherapy as well) using the mass spectrometry approach. In general, we found that CCRT induced not only a higher number of significantly changed metabolites than RT but also changes observed in both treatments were faster and stronger in the CCRT group. Moreover, statistically significant changes were not detected in sera of patients treated with one cycle of (induction) chemotherapy alone, which corresponded to the first phase of a drug administration during CCRT (i.e., sample B). Hence, the presented results indicated that the influence of CCRT on the serum metabolome was stronger than a simple additive effect of RT and ICT, which suggested synergism of the combined treatment. Such synergism putatively observed in the CCRT group could reflect the radiosensitizing effect of cisplatin, relating to its ability to inhibit DNA repair via the nonhomologous end-joining [12].

The most remarkable effect documented in this study concerned a general decrease in serum concentration of numerous metabolites observed in a course of both CCRT and RT, particularly during the first stage of treatment. The overall decrease in serum metabolite concentration, especially lipids, was already reported in patients treated with radiotherapy because of head and neck cancers [8,13] and breast [14] cancers. Moreover, the decreased levels of apolipoproteins were observed in the serum HNSCC patients during the RT [15]. Such a treatment-related decrease in levels of serum lipids could be hypothetically associated with possible activation of ketosis and cachexia in patients treated with RT and/or chemotherapy. It was shown that among the metabolites affected in HNSCC patients during the RT/CHRT treatment was 3-hydroxybutyric acid [9,16], which could be considered as an indicator of early nutritional disturbances. Noteworthy, however, in our current study, the colorimetric analysis of 3-hydroxybutyrate did not reveal any statistically significant changes related to the treatment (data not showed). The malnutrition and body mass loss are well-known factors associated with the escalation of anticancer treatment [17] and the loss of patients’ body mass was also observed in our study. However, even though the overall decrease of body mass and decrease of serum phospholipids was noted during the first stage of a treatment, no correlation between both parameters was observed for individual patients and, in contrast to the progressing loss of body mass, the level of serum phospholipids slightly increased in the second stage of a treatment. Hence, the observed changes in serum concentration of phospholipids were not directly associated with the treatment-related loss of patients’ body mass. Moreover, we observed the decrease in serum concentration of albumin that, during the first stage of treatment, correlated with the decrease of serum phospholipids. Hence, it is important to note that the reduced level of serum albumin is associated not only with a nutritional body status but also with infection, inflammation, and trauma [18]. Another factor that might be indicative of inflammation is the ratio of lysoPCs to PCs. Cellular lysoPCs emerge from phospholipase A2-catalyzed decomposition of PCs and subsequently stimulate prostaglandin synthesis, which may lead to the intensification of inflammatory reactions. Increased plasma lysoPC/PC ratio was observed in osteoarthritis that is known as an inflammatory condition [19]. Moreover, an increased lysoPC/PC ratio was detected in the serum of mice that undergo the whole body irradiation [20]. Our study, however, did not reveal any statistically significant change of lysoPC/PC ratio during the treatment. Hence, the functional association between observed changes in serum phospholipid levels and treatment-related nutrition and/or inflammation conditions remain to be validated and clarified in further studies.

Our study documented significant changes in serum levels of choline-containing phospholipids that could be related to anticancer treatment. The main source of choline is absorption from the food, however, to a lower extent, it can also be produced via the hepatic phosphatidylethanolamine N-methyltransferase pathway in the liver or through hydrolysis of membrane lipids (PCs and SMs) [21]. As a consequence, choline and choline-based compounds are constantly transformed into each other [22]. Therefore, observed changes in serum levels of choline-containing phospholipids (PCs, lysoPCs, and SMs) apparently reflect membrane regeneration processes and signal-transduction pathways associated with treatment-induced damage of cellular and tissue components.

Radiotherapy and chemo-radiotherapy resulted in the acute radiation toxicity that built-up throughout the treatment. The degree of acute reactions is affected both by the individual patient’s predisposition and escalation of treatment, yet its mechanism always involves inflammation. Inflammation, both at the local and systemic levels, preceded the occurrence of oral mucositis in HNSCC patients [23]. Acute mucositis could worsen dysphagia resulting in the accelerated loss of body mass and cachexia [24]. In our study, acute radiation toxicity increased in the response to both RT and CCRT. However, the degree of response seemed higher in the latter group (though the difference was not statistically significant because of the small size of compared groups), which might be mirrored as stronger treatment-related changes observed at the level of serum metabolites. Nevertheless, in both patient groups, treatment-related changes observed at the level of serum metabolome putatively preceded functional and morphological reactions that were barely detectable after the first stage of treatment. Hence, one should conclude that mass-spectrometry-based serum metabolomics might deliver fast and accurate information about systemic response to radio- and/or chemo-radiotherapy. 

## 4. Materials and Methods 

### 4.1. Characteristics of the Patient Group

Forty-seven patients (all Caucasians) with locally advanced HNSCC (no distant metastases) were enrolled in this study. Patients were divided into three groups depending on the received treatment: concurrent chemo-radiotherapy (CCRT; 16 patients), radiotherapy alone (RT; 18 patients), and induction chemotherapy (ICT; 13 patients); no surgery was applied before the treatment (there were more males than females in all compared groups, which corresponded the general gender ratio among the HNSCC patients). Clinical characteristics of patients and details of treatment are presented in Table 3. The study was approved by the appropriate local Ethics Committee (MSCI; approval no. 1/2016) and all participants provided informed consent indicating their conscious and voluntary participation. Acute toxicity (acute radiation sequel, ARS) was evaluated using a multiparametric scoring system based on morphological and functional factors [25]. The ARS system is based on the existing rules of the Common Toxicity Criteria of Adverse Event (CTCAE) scale and considers all symptoms related to the irradiated fields and affected functions collectively. Morphological evaluation was done by routine otorhinolaryngology examination of six anatomical areas, while the functional assessment was based on patients’ anamnesis.

### 4.2. Material Collection

Blood samples were collected before the start of treatment (sample A), about 2 weeks after the start of RT and CCRT or after the end of ICT (sample B), and directly after the end RT and CCRT (sample C), as described in Figure 1. Blood samples (5 mL) were collected into BD Vacutainer Tubes and incubated for 30 min at room temperature. Next, they were centrifuged at 1000× *g* for 10 min to remove clots. The resulting sera were portioned, then frozen and stored at −85 °C.

### 4.3. LC-MS Targeted Metabolomics

Serum samples (10 µL) were analyzed by a targeted quantitative approach using a combined direct flow injection and liquid chromatography (LC) tandem mass spectrometry (MS/MS) assay using the Absolute IDQ p180 kit (Biocrates Life Sciences AG) according to the manufacturer’s protocol. This strategy hypothetically allows simultaneous quantification of 185 metabolites: 40 amino acids and biogenic amines, 40 acylcarnitines, 90 glycerophospholipids, and 15 sphingomyelins. The method combines derivatization and extraction of analytes with the selective mass-spectrometric detection using multiple reaction monitoring and integrated isotope-labeled internal standards absolute quantification. Mass spectrometry analyses were carried out on a TSQ Vantage EMR (Thermo SCIENTIFIC) equipped with a Surveyor HPLC system (Thermo SCIENTIFIC) using an Agilent Zorbax Eclipse XDB-C18 (3.5 μm) 3.0 × 100 mm column and controlled by Xcalibur 2.1. software. The acquired data were processed using Xcalibur 2.1. and MetIDQ (Biocrates Life Sciences AG) software. Data normalization was performed according to the kit manufacturer’s protocol based on the standard serum sample. Concentrations of all metabolites were calculated in μM.

### 4.4. Analysis of Selected Amino Acids and Their Derivatives

To establish the concentration of 20 biogenic amino acids and their selected derivatives (ADMA, SDMA, NMMA, creatinine, taurine, 1-methyl-histidine, betaine, and gamma-aminobutyric acid), an aliquot of 20 µL serum was extracted with acetonitrile in proportion 1:2.4 with the addition of the internal standard solution (2-chloroadenosine). Resulting extracts were analyzed using high-performance liquid chromatography-mass spectrometry (LC/MS). The analysis was performed on a Surveyor HPLC system coupled to a TSQ Vantage Triple-Stage Quadrupole mass spectrometer according to the procedure described in detail elsewhere [26]. Chromatographic separation was performed using a 50 × 2 mm Synergi Hydro-RP 100 column (with a 2.5 µm particle size). Individual amino acids and IS were identified and confirmed by the similarity of ion masses, fragmentation pattern, and chromatographic retention time.

### 4.5. Albumin and 3-Hydroxybutyrate Detection

The concentration of serum albumin was measured by immune nephelometry using an Atellica NEPH 630 System (Siemens Healthineers) according to the manufacturer’s protocol. The concentration of serum β-HB (3-hydroxybutyrate) was measured using β-Hydroxybutyrate (Ketone Body) Colorimetric Assay Kit (Cayman Chemical) according to the manufacturer’s protocol.

### 4.6. Statistical and Bioinformatic Analyses

All analyses were performed using the R Statistical Software (v. 3.5.2). Quantitative analysis of serum using Absolute IDQ p180 kit delivered information about 185 metabolites. Metabolites whose amounts were measured in less than 80% analyzed samples were removed from the analysis, which left 149 compounds. There were 35 metabolites left that had some missed values (i.e., zero or infinity), which were imputed using statistical approaches. Zeros were filled by numbers randomly chosen between zero and the lowest measured concentration of a given metabolite. Infinities were filled by numbers randomly chosen from the end of Gaussian distribution of a given metabolite above the highest measured concentration. Principal Component Analysis (PCA) was performed based on 149 metabolites separately for samples A, B, and C independently. To detect differences between samples A, B, and C (separately for the RT and CCRT groups), an ANOVA Friedman test for dependent samples with correction for multiple testing was performed. To calculate pairwise comparisons between time points, the Nemenyi post-hoc test was used. Pairwise comparisons between samples A and B in the ICT group were calculated using the Wilcoxon pairwise test corrected for multiple testing. In order to identify specificity and differences in response among treatments, together with the standard approach where the expressions for A, B, and C time points are compared among the treatment groups (ANOVA with repeated measures algorithm), the individual responses defined as B-A, C-A, C-B between treatments were also considered. In the case of ICT protocol (B-A difference only), the Kruskal–Wallis test corrected for multiple testing was applied. Nemenyi post-hoc test was used to determine which treatments are different in a paired comparison. In the case of C-A and C-B differences, the Mann–Whitney test corrected for multiple testing was used. All calculation was considered statistically significant if the Benjamini–Hochberg false discovery rate (FDR) was below 0.05. All statistical analyses were performed using the stats package, graphs were created using the ggpubr package. 

## 5. Conclusions

Concurrent chemo-radiotherapy and radiotherapy alone induced systemic molecular changes that could be observed at the level of serum metabolome, which was exemplified by the reduced concentrations of choline-containing phospholipids. Noteworthy, the combined treatment resulted in faster and stronger effects. However, chemotherapy alone did not result in significant serum metabolome changes, which suggested that the effects of concurrent chemo-radiotherapy could not be attributed to the simple additive effects of radiotherapy alone and chemotherapy alone. We concluded that different molecular changes were measured at the level of serum metabolome in response to treatment modalities hypothetically associated with different risk of toxicity. 

## Figures and Tables

**Figure 1 metabolites-10-00060-f001:**
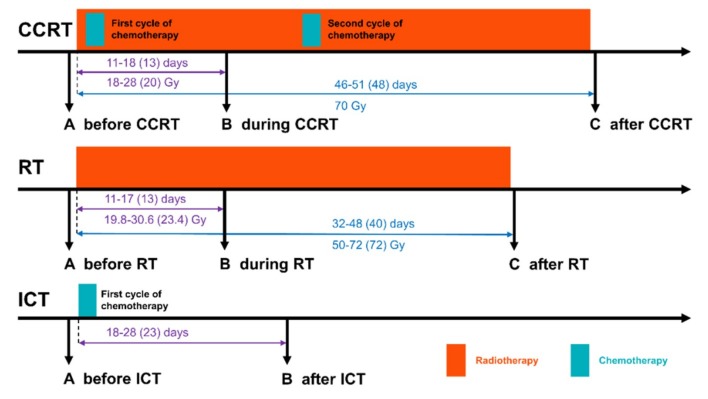
Schematic representation of treatment schemes included in the study: CCRT (concurrent chemo-radiotherapy), RT (radiotherapy), and ICT (induction chemotherapy). Marked are numbers of days and received doses between the start of the treatment and sample B or sample C (minimum-maximum and median in the parentheses).

**Figure 2 metabolites-10-00060-f002:**
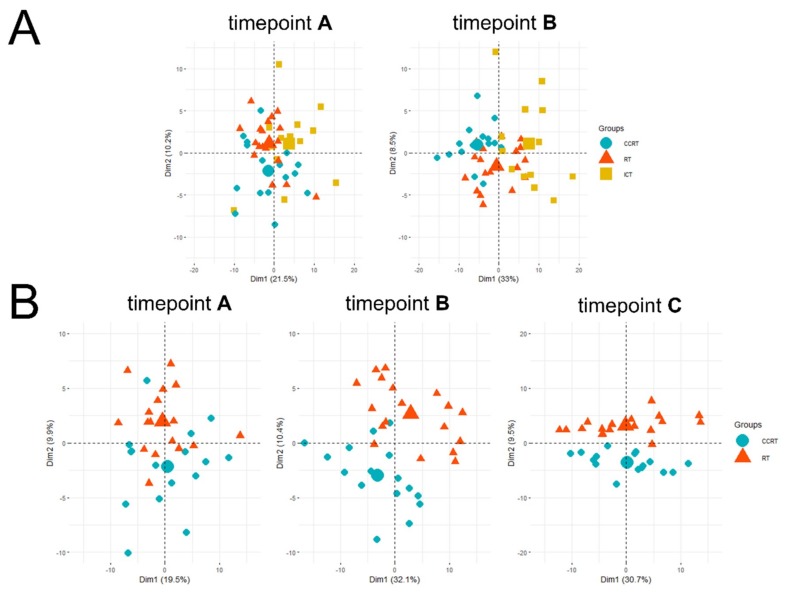
Principal component analysis (PCA) of the sample similarity. Panel (**A**) shows samples A and samples B collected in all three treatment modalities. Panel (**B**) shows three subsequent time points A, B, and C that correspond to samples collected before, during, and after the end of CCRT and RT treatments, respectively. Small marks denote individual patients, while bigger marks are the mean points of each treatment group. CCRT is labeled by blue dots, RT—orange triangles, and ICT—yellow squares.

**Figure 3 metabolites-10-00060-f003:**
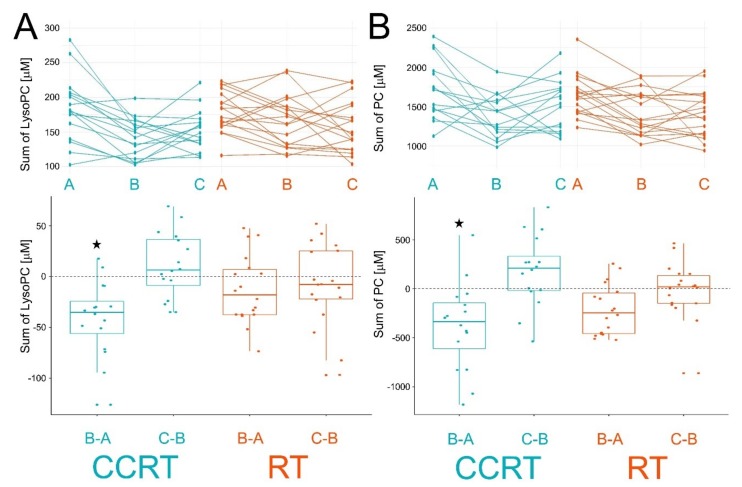
Individual changes of total lysoPC (Panel A) and PC (Panel B) levels between consecutive samples. Showed are time courses of individual levels (upper graphs; samples A, B, and C) and differences between consecutive samples (bottom graphs; changes B-A and C-B). Boxplots show minimum, lower quartile, median, upper quartile, and maximum values; dots represent individual differences; significant change (FDR < 0.05) is marked with asterisks. CCRT samples are marked in blue and RT samples are marked in orange.

**Figure 4 metabolites-10-00060-f004:**
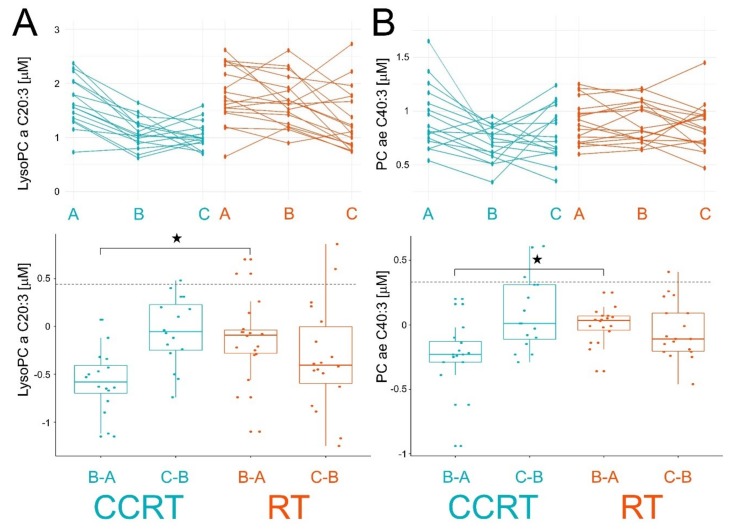
Individual changes in levels of lysoPC a C20:3 (Panel A) and PC ae C40:3 (Panel B) between consecutive samples. Showed are time courses of individual levels (upper graphs; samples A, B, and C) and differences between consecutive samples (bottom graphs; changes B-A and C-B). Boxplots show minimum, lower quartile, median, upper quartile, and maximum values; dots represent individual differences; significant change (FDR < 0.05) is marked with asterisks. CCRT samples are marked in blue and RT samples are marked in orange.

**Figure 5 metabolites-10-00060-f005:**
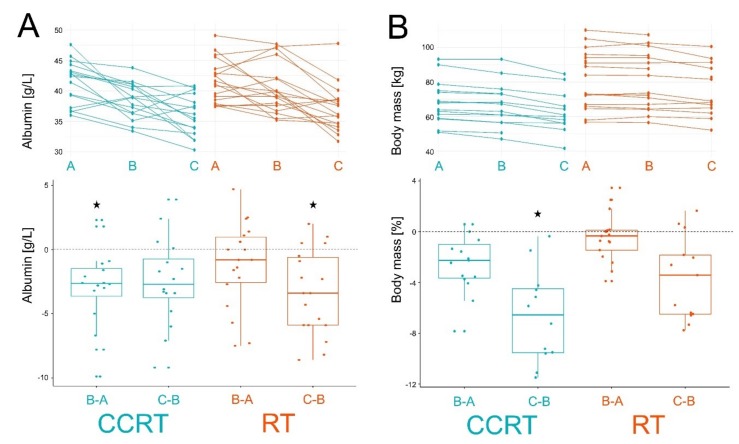
Treatment-related changes in serum albumin panel (**A**) and body mass panel (**B**). Depicted are time courses of individual levels (upper graphs; samples A, B, and C) and differences between consecutive samples (bottom graphs; changes B-A and C-B). Boxplots show minimum, lower quartile, median, upper quartile, and maximum values; dots represent individual differences; significant change (FDR < 0.05) is marked with asterisks. CCRT samples are marked in green and RT samples are marked in orange.

**Figure 6 metabolites-10-00060-f006:**
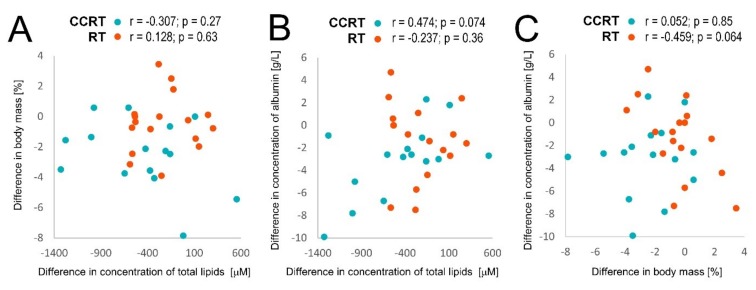
Correlations between individual treatment-related changes of total phospholipid concentration (PC, lysoPC, and SM), serum albumin concentration, and body mass during the first stage of a treatment (the B-A difference) in CCRT and RT samples marked in blue and orange, respectively. Denoted are the correlation coefficient (r) and its p-value for each patients’ group. Showed are: correlations between changes in body mass and concentration of total serum lipids (**A**), correlations between changes in concentration of serum albumin and total serum lipids (**B**), correlations between changes in concentration of serum albumin and body mass (**C**).

**Figure 7 metabolites-10-00060-f007:**
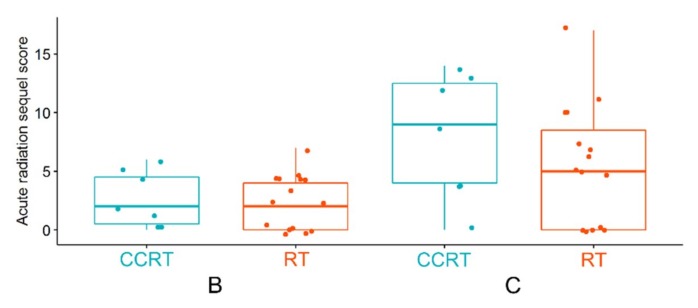
Acute radiation sequel (ARS) estimated based on morphological and functional reactions in patients from RT and CCRT groups during a treatment (time point B) and after the end of treatment (time point C).

**Table 1 metabolites-10-00060-t001:** Treatment-induced changes in serum metabolites. The following columns show their levels in the subsequent samples (µM; mean values in samples A, B, and C) and changes between them (change B-A and C-B, mean values ± standard deviation; asterisks mark statistically significant changes - false discovery rate (FDR) > 0.05). Patterns of changes between A vs. B and B vs. C are also presented (empty cells correspond to no significant change in both stages of treatment; i.e., pattern A = B = C).

Patient Groups	CCRT					RT					Pattern of Changes	
	Level (Mean Value)			Change (Mean Value ± S.D.)		Level (Mean Value)			Change (Mean Value ± S.D.)			
Metabolites	A	B	C	B-A	C-B	A	B	C	B-A	C-B	CCRT	RT
ac C0	47.36	42.61	41.17	−4.75 ± 11.42	−1.44 ± 13.61	43.04	43.44	36.34	0.4 ± 9.37	−7.1 * ± 9.95		A = B > C
ac C16	0.11	0.08	0.10	−0.03 * ± 0.04	0.02 ± 0.04	0.11	0.09	0.10	−0.01 ± 0.05	0.01 ± 0.06	A > B = C	
ac C18:1	0.19	0.14	0.17	−0.05 * ± 0.08	0.03 ± 0.1	0.15	0.12	0.14	−0.03 ± 0.05	0.02 ± 0.04	A > B = C	
Cit	36.99	27.89	26.86	−9.1 * ± 8.48	−1.03 ± 9.09	36.12	33.49	27.69	−2.63 ± 8.29	−5.8 ± 6.36	A > B = C	
His	102.29	80.43	75.00	−21.86 * ± 11.88	−5.43 ± 16.09	104.68	95.38	83.02	−9.29 ± 11.91	−12.37 ± 18.83	A > B = C	
Pro	212.13	157.87	162.69	−54.26 * ± 38.47	4.82 ± 47.2	221.00	211.28	189.72	−9.72 ± 46.7	−21.56 ± 49.33	A > B = C	
kynurenine	3.01	2.51	2.14	−0.5 * ± 0.79	−0.36 ± 0.75	3.20	3.13	2.38	−0.07 ± 0.83	−0.75 * ± 0.61	A > B = C	A = B > C
lysoPC a C16:0	103.65	83.23	88.67	−20.42 * ± 21.36	5.44 ± 16.54	99.78	92.29	89.77	−7.49 ± 20.07	−2.52 ± 21.64	A > B = C	
lysoPC a C16:1	2.77	1.86	1.85	−0.9 * ± 0.87	−0.01 ± 0.42	2.75	2.43	2.15	−0.32 ± 0.89	−0.28 ± 0.82	A > B = C	
lysoPC a C18:0	28.62	21.61	20.34	−7.01 * ± 7.93	−1.28 ± 7.32	28.64	26.73	22.12	−1.92 ± 6.27	−4.61 ± 7.18	A > B = C	
lysoPC a C18:1	18.95	13.84	17.20	−5.11 * ± 4.75	3.36 ± 5.9	17.77	16.30	16.22	−1.47 ± 4.02	−0.08 ± 5.29	A > B = C	
lysoPC a C18:2	17.48	11.63	14.20	−5.85 * ± 5.11	2.57 ± 4.63	16.22	15.22	14.44	−1 ± 4.33	−0.78 ± 6.49	A > B = C	
lysoPC a C20:3	1.66	1.09	1.04	−0.57 * ± 0.33	−0.05 ± 0.36	1.79	1.66	1.33	−0.13 ± 0.42	−0.33 ± 0.56	A > B = C	
lysoPC a C20:4	5.05	4.08	4.97	−0.97 ± 1.33	0.89 * ± 1.37	5.90	5.58	5.86	−0.32 ± 1.03	0.29 ± 2.01	A = B<C	
PC aa C32:0	14.90	13.68	17.64	−1.23 ± 4.54	3.97 * ± 4.35	12.96	12.11	13.89	−0.85 ± 2.66	1.78 ± 2.85	A = B<C	
PC aa C32:1	22.08	15.32	15.70	−6.76 ± 10.39	0.38 ± 6.35	16.55	13.13	12.94	−3.43 * ± 4.82	−0.19 ± 7.21		A > B = C
PC aa C32:2	2.21	1.66	2.07	−0.55 ± 0.64	0.42 ± 0.72	2.34	2.33	0.99	−0.01 ± 1.55	−1.34 * ± 1.71		A = B > C
PC aa C32:3	0.36	0.29	0.36	−0.07 * ± 0.08	0.07 * ± 0.09	0.37	0.36	0.36	−0.01 ± 0.08	−0.01 ± 0.1	A > B<C	
PC aa C34:2	332.25	247.81	308.13	−66.75 * ± 100.8	46.19 * ± 73.68	295.44	245.61	265.72	−41 * ± 49.04	28.33 ± 59.29	A > B<C	A > B = C
PC aa C34:4	1.20	0.82	0.81	−0.38 ± 0.49	−0.01 ± 0.43	1.14	1.03	0.71	−0.11 ± 0.25	−0.32 * ± 0.45		A = B > C
PC aa C36:2	188.38	137.85	148.95	−50.53 * ± 52.78	11.1 ± 49.89	181.61	155.31	136.73	−26.31 ± 23.73	−18.57 ± 36.15	A > B = C	
PC aa C36:3	111.54	75.59	76.28	−35.94 * ± 34.14	0.68 ± 29.13	109.86	93.58	78.48	−16.28 * ± 19.03	−15.1 ± 25.9	A > B = C	A > B = C
PC aa C36:6	0.76	0.54	0.51	−0.22 ± 0.3	−0.03 ± 0.27	0.72	0.73	0.50	0.01 ± 0.29	−0.22 * ± 0.28		A = B > C
PC aa C38:3	45.04	34.30	32.19	−0.5 * ± 0.58	0.19 ± 0.83	47.89	41.80	33.03	−0.23 ± 0.79	−0.14 * ± 0.93	A > B = C	A = B > C
PC aa C40:4	3.49	2.66	2.63	−0.83 ± 1.06	−0.03 ± 0.86	3.40	2.89	2.49	−0.5 * ± 0.67	−0.41 ± 0.79		A > B = C
PC aa C40:5	10.28	7.57	6.91	−2.71 * ± 3.4	−0.65 ± 2.77	10.26	9.26	7.74	−1 ± 2.69	−1.52 * ± 2.99	A > B = C	A = B > C
PC aa C42:1	0.28	0.22	0.21	−0.06 * ± 0.08	−0.01 ± 0.09	0.27	0.23	0.23	−0.04 ± 0.12	0 ± 0.11	A > B = C	
PC aa C42:6	0.38	0.28	0.30	−0.1 ± 0.12	0.02 ± 0.11	0.44	0.42	0.32	−0.03 ± 0.15	−0.1 * ± 0.13		A = B > C
PC ae C34:3	5.59	4.05	6.53	−2.48 * ± 2.23	1.67 * ± 2.68	5.67	5.02	5.44	−1.44 ± 2.08	−0.29 ± 2.74	A > B<C	
PC ae C36:2	9.24	7.08	8.77	−2.15 * ± 1.71	1.69 ± 3.1	10.54	9.65	9.24	−0.89 ± 2.23	−0.41 ± 2.63	A > B = C	
PC ae C36:3	5.62	3.54	4.15	−2.08 * ± 1.36	0.61 ± 1.69	6.61	5.57	5.07	−1.04 * ± 1.54	−0.51 ± 1.77	A > B = C	A > B = C
PC ae C36:4	15.63	11.42	12.08	−4.21 * ± 3.5	0.66 ± 3.82	16.73	14.19	13.17	−2.54 ± 4.14	−1.02 ± 4	A > B = C	
PC ae C38:2	1.57	1.13	1.10	−0.44 * ± 0.45	−0.02 ± 0.54	1.43	1.25	1.16	−0.18 ± 0.26	−0.08 ± 0.44	A > B = C	
PC ae C38:3	3.33	2.39	2.70	−0.94 * ± 0.8	0.31 ± 0.77	3.54	3.28	2.85	−0.26 ± 0.69	−0.44 ± 0.93	A > B = C	
PC ae C38:4	11.26	8.77	9.64	−2.5 * ± 2.28	0.87 ± 3.17	12.62	10.97	10.33	−1.65 * ± 2.13	−0.64 ± 2.06	A > B = C	A > B = C
PC ae C38:5	16.56	12.74	14.99	−3.81 * ± 3.23	2.25 ± 4.54	18.93	16.33	16.13	−2.6 * ± 3.67	−0.2 ± 3.84	A > B = C	A > B = C
PC ae C40:3	0.94	0.71	0.79	−0.24 * ± 0.27	0.09 ± 0.29	0.90	0.90	0.85	0 ± 0.14	−0.05 ± 0.22	A > B = C	
PC ae C42:2	0.57	0.43	0.40	−0.14 * ± 0.15	−0.03 ± 0.17	0.59	0.55	0.49	−0.05 ± 0.15	−0.06 ± 0.17	A > B = C	
SM (OH) C14:1	6.23	5.64	7.14	−0.59 ± 1.72	1.5 * ± 1.89	5.28	5.23	5.94	−0.05 ± 1.01	0.72 ± 1	A = B<C	
SM (OH) C16:1	3.20	3.17	4.15	−0.03 ± 0.67	0.98 * ± 1.2	3.17	3.22	3.87	0.04 ± 0.89	0.66 * ± 0.8	A = B<C	A = B<C
SM (OH) C22:1	10.50	8.13	8.73	−2.38 * ± 2.18	0.6 ± 3.3	9.72	8.94	8.64	−0.77 ± 2.3	−0.31 ± 2.57	A > B = C	
SM C16:1	16.58	14.85	19.13	−1.73 ± 4.34	4.27 * ± 4.83	15.71	14.42	16.54	−1.28 ± 3.06	2.12 ± 2.83	A = B<C	
SM C18:0	29.49	29.20	39.38	−0.29 ± 9.17	10.18 * ± 11.71	26.62	25.94	32.77	−0.68 ± 7.49	6.82 * ± 5.69	A = B<C	A = B<C
SM C24:0	21.65	15.79	16.06	−5.86 * ± 4.96	0.27 ± 5.79	18.77	16.57	15.49	−2.21 ± 3.86	−1.08 ± 3.59	A > B = C	
SM C24:1	63.66	56.36	67.73	−7.31 ± 14.77	11.37 ± 16.82	58.63	55.34	65.98	−3.29 ± 14.52	10.64 * ± 9.45		A = B<C

Abbreviations used: ac—acyl carnitine; a—acyl; aa—diacyl; ae—acyl-alkyl; lysoPC—lysophosphatidylcholine; PC—phosphatidylcholine; SM—sphingomyelin; SM (OH) —hydroxysphingomyelin. Listed lipids represent groups of isomers that cannot be separated using the implemented LC-MS approach. Names of compounds reflect numbers of carbon atoms and double bonds in fatty acid residues. IDs of specific lipid species included in each isomer group are listed in the Supplementary File Appendix A.

**Table 2 metabolites-10-00060-t002:** Metabolites affected by CCRT and RT. Presented are metabolites that changed their levels significantly (FDR < 0.05) between specified time points in both treatment modalities.

**Metabolites**	**RT**	**CCRT**
	**Ratio B/A**	
PC aa C34:2	0.847	0.809
PC aa C36:3	0.854	0.713
PC ae C36:3	0.854	0.652
PC ae C38:4	0.876	0.779
PC ae C38:5	0.873	0.776
	**Ratio C/B**	
SM (OH) C16:1	1.245	1.387
SM C18:0	1.301	1.421
	**Ratio C/A**	
C0	0.869	0.895
His	0.797	0.742
Pro	0.868	0.792
Kynurenine	0.775	0.756
lysoPC a C20:3	0.787	0.668
PC aa C36:3	0.714	0.719
PC aa C38:3	0.687	0.733
PC aa C40:5	0.774	0.720
PC ae C36:3	0.783	0.780
PC ae C38:3	0.815	0.843
PC ae C38:4	0.835	0.872
SM (OH) C16:1	1.270	1.322
SM C18:0	1.289	1.347
SM C24:0	0.825	0.756

**Table 3 metabolites-10-00060-t003:** Summary of clinical characteristics of the patients.

Characteristics	CCRT	RT	ICT
Number of patients	16	18	13
Age [years] (median)	49–75 (58)	55–77 (63)	46–80 (63)
Gender: male/female	13/3	12/6	11/2
**Tumor localization**			
Oral cavity	0	2	3
Tonsil	2	3	3
Pharynx	7	1	7
Larynx	7	12	0
**TNM staging**			
T1	0	1	0
T2	4	11	1
T3	7	6	6
T4	5	0	6
N0	6	16	1
N1	2	1	1
N2	8	1	7
N3	0	0	4
**Radiotherapy**			
Total dose [Gy] (median)	70	50-72 (72)	-
7x1,8Gy/week	0	10	-
5x2Gy/week	16	2	-
5x2,2Gy/week	0	4	-
5x2,5Gy/week	0	2	-
**Chemotherapy**			
Cisplatin (100 mg/m^2^)	16	-	5
Cisplatin (100 mg/m^2^) + 5-fluorouracil (800 mg/m^2^)	0	-	8

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
