# Peer review of "Systemic Effects of Radiotherapy and Concurrent Chemo-Radiotherapy in Head and Neck Cancer Patients—Comparison of Serum Metabolome Profiles"

_metabolites, 2020, doi:10.3390/metabo10020060_

Round 1
Reviewer 1 Report
I agree that metabolomics is an extensively developing and very interesting field, but I have some major issues with this work:
You conclude that the molecular response measured at the level of serum metabolome preceded acute radiation toxicity observed at the morphological and functional levels. Nevertheless, any direct association between Acute Radiation Sequel score (ARS) and levels of studied metabolites was not shown. You discussed only the differences between RT and CCRT which is rather screening than clinically useful approach. In my opinion, any proof of an association between metabolite levels and high or low ARS could significantly improve this work. I do not understand, why the levels of studied metabolites are shown as B-A, C-B (Figs 3,4,5) and not as levels in point A, B, and C. Please explain or change the presentation of your data because this presentation is a bit confusing. (B-A means B minus A level?)Minor issues:
There is a typing error; line 83 (In general, there were 149 metabolites detec16ted and quantified in analyzed samples). Both tables are numbered as Table 1. The sentence (lines 170 and 171) has no sense.Author Response
Please see the attachment.

Reviewer 2 Report
Jelonek et al. conducted a comprehensive study comparing the serum metabolome profiles of the head and neck cancer patients that went through different treatment protocols. I believe that this study is worthy of publication. Yet, I have several comments that hopefully will improve the overall merit of the work. Please refer to the below comments for more information.
Section 4.6, the authors made a comprehensive missing value imputation approach. However, I would ask if any further data treatment was done regarding (1) the relative standard deviation of detected metabolites in quality control samples (if available); (2) any data scaling or transformation applied prior to the visualization with PCA. Similarly, the authors did a careful univariate pairwise statistical analysis. However, no supervised multivariate was done. Please kindly explain your rationale. The annotation of the metabolites, especially lipids, was not satisfactory. I understand that the authors followed the p180 kit for the lipid annotation. However, a robust notation system following LIPID MAPS or Liebisch et al. (10.1194/jlr.M033506) or any other systems of choice is strongly recommended. Line 98-112: Since the authors applied a targeted approach, it is not convincing to conclude that “phospholipids were the major target of treatment-induced changes observed in patients’ sera”. The statement should be revisited to avoid misleading comprehension of the readers. I guess one of the ways is to add “Among targeted metabolites, phospholipids appeared to be significantly altered compared with others.” I am also not convinced with the statement given “data not showed” here. As a general rule, if we do not present the data, we do not discuss it. I would recommend the authors to express some limitations of the study, including the sampling and pre-analytical steps. For instance, the research population is heavily biased in the male gender. Also, the serum was not immediately quenched using liquid nitrogen. The differences observed were slight but no multivariate modeling with proper cross-validation was applied for a better estimate of the possibility to use serum metabolome to “predict” systemic response to the chemo-radiotherapy. In fact, the comparison of the metabolome before-treatment and after-treatment is suitable to describe “what might change”. It is also OK, although not perfect, to compare the similarity and the differences regarding the “patterns of metabolome alterations” of different routes of treatments. Nonetheless, it is not suitable for the purposes of predicting “what would happen to the patient if we applied this treatment given their baseline metabolome”. I strongly recommend the authors to write down the package in R they employed to perform the statistics and visualize the results. Line 329, the term “molecular weights” should be better changed to “ion mass” or “mass-to-charge ratio” (m/z) since mass spectrometer does not monitor “molecular weight”.
Round 2
Reviewer 1 Report
I think your following conclusion: (Moreover, the systemic response to treatment measured at the level of serum metabolome was observed faster than acute radiation toxicity that was estimated based on morphological and functional parameters) is not sufficiently supported by the results. Please remove this conclusion from the abstract and the conclusion part.
Reviewer 2 Report
The authors significantly improved the quality of the manuscript. I hereby recommend it to be published on Metabolites.
Author Response
Thank you for the approval of the improved manuscript.